# Canine Socialisation: A Narrative Systematic Review

**DOI:** 10.3390/ani12212895

**Published:** 2022-10-22

**Authors:** Victoria McEvoy, Uri Baqueiro Espinosa, Andrew Crump, Gareth Arnott

**Affiliations:** 1Institute for Global Food Security, School of Biological Sciences, Queen’s University Belfast, 19 Chlorine Gardens, Belfast BT9 5DL, UK; ubaqueiroespinosa01@qub.ac.uk (U.B.E.); g.arnott@qub.ac.uk (G.A.); 2Centre for Philosophy of Natural and Social Science, London School of Economics and Political Science, Houghton St., Holborn, London WC2A 2AE, UK; andrewcrump94@gmail.com

**Keywords:** dog, puppy, *Canis familiaris*, socialisation, socialisation period, early life

## Abstract

**Simple Summary:**

New dog owners are given a plethora of advice on how to socialise their puppy, but such advice is often outdated and based on very few experimental studies. The resulting inadequate socialisation can lead to behavioural problems in adult dogs. This review aims to describe all relevant literature regarding canine socialisation. Many of the 29 studies identified were retrospective owner-filled questionnaires, which are susceptible to bias. Few modern studies experimentally investigated the effects of different socialisation methods. We, therefore, recommend studies on the minimum necessary level of socialisation and breed differences in the optimum timing for socialisation. We hope this future research helps owners and breeders to produce well-adjusted dogs.

**Abstract:**

There are over 10 million pet dogs in the UK alone, and they have become a member of modern human families. If not properly socialised as puppies, dogs have a higher risk of problematic behaviours during adulthood, yet socialisation studies are lacking. Much of the experimental research was carried out at least 50 years ago, and the importance of socialisation was demonstrated so clearly that further studies with unsocialised controls would be deemed unethical. In this review, the aim was to evaluate all literature relevant to canine socialisation. This review used PRISMA-P guidelines to identify 29 studies: 14 were questionnaire-based studies (two of which also had a testing element), 15 included some form of experimental manipulation relating to socialisation, and one was a purely observational study. Based on this literature review, we recommend future research into minimum necessary socialisation levels, as well as breed differences in the timing of effective socialisation. Such studies will help owners and breeders produce well-adjusted adult dogs.

## 1. Introduction

The European Pet Food Industry (FEDIAF) estimate that there are 92.9 million pet dogs in Europe, with 10–13 million reported in the UK [1,2,3]. Two-and-a-half million dogs were acquired in the UK between March 2020 and March 2022. These came from a range of sources. According to the People’s Dispensary for Sick Animals (PDSA) 2022 Animal Wellbeing (PAW) Report, 32% of UK dog owners sourced their pet from a breeder, 14% from a rescue/rehoming centre, and 23% from a private seller. With the increasing demand for pet dogs and diversity of rearing environments, appropriate early-life socialisation is becoming ever more important.

Early experiences, whether positive or negative, can profoundly affect behaviour later in life. Many studies have investigated the influence of these early experiences in humans, including how they impact adult psychopathologies, such as depression and schizophrenia [4,5,6,7]. Early experiences also affect adults in many other species, including cats, [8], pigs [9,10,11,12,13], foxes [14,15,16,17,18], mice [19], and other farm animals, such as cattle and sheep [20,21,22,23]. For example, giving kittens additional socialisation led to owners reporting significantly higher emotional support from their cats later in life. Compared to kittens that did not receive additional socialisation, socialised kittens also exhibited less fear-related behaviours towards humans at one year old [24]. 

Positive early experiences during the ‘critical’ or what are now, due to the plasticity of behaviour and preferences acquired during these periods, referred to as ‘sensitive’ periods of development are crucial to create well-adjusted adult dogs able to cope in their environment. There are currently six defined sensitive periods in early canine development: (1) the prenatal period (9 week gestation period), (2) the neonatal period (birth to 2 weeks of age), (3) the transition period (2–3 weeks of age), (4) the socialisation period (3–12 weeks of age), (5) the juvenile period (12 weeks to 6 months of age), and (6) the pubertal period (7–24 months) [5,25,26]. Scott and Fuller (1965) [27] originally described the socialisation period as a “critical period” in the formation of primary social relationships or attachments [27]. During this time, puppies not only show pro-social tendencies in both intra- and interspecific interactions, but also a reduction in fear/avoidance tendencies in novel situations. Between 3–5 weeks, puppies show a higher tendency to approach an unfamiliar person; this tendency subsequently declines. Scott and Fuller (1965) [27], therefore, concluded that primary socialisation runs from 3–12 weeks after birth [27]. They also concluded that puppies become strongly attached to a place during the same time period, a phenomenon they termed ‘localisation’. They state that ‘the results of socialisation and localisation are so similar that [they] wonder whether they may represent the same process applied to different objects. The terms socialisation and localisation as a result have become synonymous in the study of the socialisation period even though they may represent different developmental processes. As a result of these conclusions, they advised that puppies should be introduced to stimuli and conditions they are likely to encounter as adults, in order to produce well-balanced and well-adjusted adult dogs able to cope with novelty [25,28,29]. Not exposing puppies to stimulating and positive experiences during this period can lead to adult behavioural problems, such as separation anxiety and aggression [30]. These behavioural problems are the some of the main reasons pet dogs are surrendered to shelters or euthanised. One study analysed the incidence of dog bite admissions between 1998–2018 and found a rise from 6.34 to 14.9 admissions per 100,000 people, the cost of which peaked at £25.1 million in admission costs and £45.7 million in emergency attendance costs, presenting a public health issue and economic costs [29,31,32,33,34].

The socialisation period includes the time when commercially-bred puppies are in the breeding facility, as homing happens from 8 weeks onwards. Responsibility for proper exposure to age-appropriate socialisation, therefore, starts with the breeder [29]. In many European countries, breeders are legally obliged to socialise dogs during the socialisation period. The UK, for instance, requires commercial breeders to carry out a socialisation program on all puppies before 8 weeks. However, the specifics of these programs are not written in the legislation and are, therefore, created at the breeders’ discretion. Although local authorities must approve breeders’ socialisation programs, there is no enforcement to ensure they are carried out [35]. Moreover, the largest puppy trade network in western Europe imports puppies from Hungary and Slovakia [36], which have no guidelines regarding dog breeding or socialisation [29,37,38]. 

Inadequate socialisation is an increasingly pressing issue, and Dendoncker et al. (2019) [39] report that more puppies are being bred in large commercial breeding establishments. Dogs sourced from these facilities appear to express more adverse behaviours as adults (e.g., aggression and separation anxiety [40]). A review identified only seven studies investigating adult behaviours of dogs suspected to have been sourced from commercial breeding establishments. The main behavioural disorders reported in these dogs displayed increased fear, aggression, anxiety, and separation-related behaviours, as well as attention-seeking behaviours and heightened sensitivity to touch [30]. The COVID-19 pandemic has also been reported to reduce levels of puppy socialisation. For example, Brand et al. (2022) [41] found that “Pandemic Puppies” were less likely to have attended puppy training classes or received exposure to people from outside the home before 16 weeks of age. This raises concerns for the future welfare of these puppies and highlights the need for more research into socialisation programs [41]. 

Most socialisation programs available to the public are based on the research from the 1950s–1960s (e.g., [27,42,43,44,45]). However, given the change in people’s lifestyles and increasingly urban environments that dogs are expected to cope with, these early laboratory studies may not address all aspects of socialisation required to produce a well-adjusted adult dog. As Miklósi (2007) [46] highlighted, and Scott and Fuller (1965) [27] recognised, the 1960s studies only covered one methodological approach and certain breeds may develop slower than others [27,46]. Since Scott and Fuller (1965), few experimental studies have investigated the socialisation period in dogs, because robust experiments with non-socialised control groups would be unethical given the importance of this period on future behaviour [26,27]. Thus, there are large knowledge gaps in how to effectively socialise puppies.

In this review, we explore how socialisation from 3–12 weeks impacts dogs’ future behaviour. We use a systematic search method to ensure all relevant information was captured but, given the variability in study designs, the discussion will then take a narrative approach. We critically evaluate all published studies, including experimental, questionnaire-based, and observational methodologies. Finally, we highlight knowledge gaps and suggest future research priorities.

## 2. Materials and Methods

### 2.1. Literature Search Procedures

On 17 August 2022, we searched the Web of Science (https://www.webofknowledge.com/, accessed on 19 August 2022) database for dog studies on the socialisation period. This covered the Web of Science Core Collection (1970-present), KCI-Korean Journal Database (1980-present), MEDLINE (1950-present), and the SciELO Citation Index (1997-present). The search string was: (TS = (dog* OR pup* OR cani* NOT pupil*)) AND TS = (sociali?ation*). TS (topic search) identified articles with the search terms in the title, abstract, author list, and keywords. An asterisk indicates that the database identified words beginning with those letters. For example, dog* was used to find references relating to the word’s “dog” and “dogs”. The search returned 749 results.

To uncover any further relevant articles, we searched five relevant journals for the term “socialisation dog”: *Applied Animal Behaviour Science, Animal Behaviour, Journal of Applied Animal Welfare Science, Animal Welfare*, and *The Journal of Veterinary Behavior*. This returned an additional 107 results.

We then searched the reference lists of two relevant studies [29,47] and chapter six of *The Domestic Dog* [26].

Any further papers that have cited Scott and Fuller’s (1965) [27] original studies were also included. This yielded another 180 results, giving a total of 1036. As a final measure to ensure adequate literature coverage, experts in the field of canine early life research were consulted and any articles deemed relevant but missed by the literature searches were included under the term “expert additions”. Flow diagram depicting search process, see Figure 1.

### 2.2. Classification of Results

PRISMA-P guidelines were used to classify results [48]. The references were initially screened for relevance and duplicates. This involved reading the title and removing any obviously irrelevant references (746 in total, including duplicates), leaving only the relevant papers (239 in total). The remaining 239 papers were again screened in more detail for relevance to socialisation leaving the remaining studies to undergo quality assessment (115 in total).

### 2.3. Quality Assessment

115 studies were subjected to quality assessment using the following protocol (adapted from the REFLECT statement; [49]). Any studies not meeting the standards set out in the quality assessment protocol were discarded from the review. Factors considered were:(1)Randomization: subjects allocated randomly to treatment groups. Although this is often not specified in methods sections, papers were excluded when experimental allocation was clearly biased.(2)Control: use of a suitable control group (with the exception of questionnaire-based studies).(3)Sample size: use of a sufficiently large sample size. Studies with a sample size of less than 5 experimental units (animals) per treatment group were discarded. Festing and Altman (2002) [50] state that the degrees of freedom for the error term used to test the effect of the variable should not be less than 10 [50].(4)Statistical methods: clear account of the statistical methods used to compare groups for all outcomes, use of appropriate statistical methods, and, where applicable, use of methods to account for non-independence of study subjects.(5)Exclusion of conference abstracts and proceedings: insufficient detail and information content for critical appraisal.(6)Exclusion of review papers. Although reviews by Dietz et al. (2018) [29] and Howell et al. (2015) [51] were used to identify potentially useful papers, they were not included in the final number of relevant studies found.(7)Socialisation did not occur during the primary socialisation period (3–14 weeks). For example, Boxall (2004) [52] speaks of socialisation on dogs and its importance in laboratory animals, but the study focuses on adult dogs.

## 3. Results

After this quality assessment, 29 studies were identified as relevant. The studies are detailed in Table 1.

## 4. Discussion

### 4.1. Classic Studies

Today’s dog trainers and dog behaviour clinicians mostly use socialisation programs informed by studies from the 1950s/1960s [51]. These studies sought to determine the upper and lower boundaries of early socialisation and so define the ideal period during which puppies were most sensitive to external stimuli. The commonly accepted time frame for primary socialisation is from 3 to approximately 12 weeks of age [26,27,79]. 

Many older studies did not meet our table inclusion criteria, but we discuss them here due to their importance for the field’s development. These studies focused on restriction of early experiences and the outcomes for adult behaviour. They showed that dogs kept in restrictive rearing conditions were significantly less likely to display ‘friendly’ behaviour towards human experimenters, and generally exhibited obvious ineptitudes in coping with social situations involving other dogs or humans. These studies would not be ethical to replicate today, because depriving puppies of socialisation has long-term negative impacts, but they were carried out before the importance of early socialisation was fully understood [81,82,83]. Without this work demonstrating the implications of restricting early experiences, the importance of the research by Scott and Marston and later Scott and Fuller, would not be so important.

A preliminary observational study demonstrating the socialisation period’s importance, Scott and Marston (1950) [79], investigated the development of puppy behaviours from birth until 16 weeks. Researchers began a regime of testing at 5 weeks, which included general handling. Before this regime, the puppies were relatively timid in the presence of researchers. After, they became less timid and this corresponded with physiological changes indicating reduced stress, such as lowered heart rates. However, the boundaries of the socialisation period could not be inferred from this study as all the puppies were handled at the same age.

Like many modern studies, Pfaffenberger and Scott (1959) [42] focused on guide dog behaviour, using success in guide dog training as an outcome measure. They explored whether additional time spent in kennels before rehoming affected training success. Dogs homed at 12 weeks (*n* = 40) passed the training with an approximate 90% success rate. Those rehomed in the second week (i.e., at 14 weeks (*n* = 22)) had slightly poorer success rates, but not significantly so, while dogs retained in the kennel more than two weeks (*n* = 18) showed a highly significant reduction in the number of training successes. This indicated that rehoming before 12 weeks was significantly correlated with guide dog training success, aspects of which assess the dog’s ability to cope with novelty and distractibility. This study provided one of the first indications that the timing of early experiences influenced future behaviour [42].

Whilst earlier studies had highlighted the socialisation period’s existence, Freedman, King and Elliot (1961) [78] sought to identify its upper and lower boundaries. They investigated the age when human exposure would most effectively reduce the withdrawal response that they stated puppies display at 14 weeks. Puppies from individual litters were taken from “the field” (i.e., the arena the puppies were kept with their littermates and mother without human interaction) for a week of socialisation at 2 weeks (*n* = 6), 3 weeks (*n* = 6), 5 weeks (*n* = 7), 7 weeks (*n* = 7), and 9 weeks (*n* = 3) of age, and then returned to the field. During the week of socialisation, the puppies were played with and “cared for” for half an hour, three times a day, although the content of the play interactions and care giving was not reported. Control puppies (*n* = 5) remained in the field until the entire litter was taken indoors for final testing at 14 weeks of age. Compared to socialised puppies, control puppies tended to withdraw from human beings after 5 weeks. In a handling test between 14 and 16 weeks, the puppies who were socialised at 9 weeks, showed more attraction to the human handler, indicating that this was the best time to socialise dogs to a human. Unless socialisation occurred before 14 weeks, withdrawal reactions from humans became so intense that normal relationships could not thereafter be established. The experimenters also retested a randomly chosen control puppy after 3 months of daily interaction and found that it only showed a slight positive change in score, although the sample size for this aspect was only one individual. Although this study broadly identified the best timing of human socialisation, it should be noted that the treatment group for puppies taken from the field at 9 weeks only contained three subjects. The study is included in this discussion as the other four treatment groups satisfied the criterion, however, the treatment group with the most significant conclusions regarding the correct timing for human interaction does not meet the same criterion. This study is often touted as the definitive study in the identification of correct timing for human introduction however, given the sample size, the power of these conclusions is minimal. [78]. 

Fuller (1967) [84] claimed that puppies could be socialised with as little as two 20-minute sessions of exposure per week from 3 weeks to 12 weeks, calling into question how much socialisation a puppy really needs. Wolfe (1990) [85] described a program that achieved “adequate socialisation” of laboratory beagles with less than five minutes human social contact per week. However, both studies were based on laboratory animals and conditions and only explored socialisation to humans, rather than general socialisation to non-social stimuli. The adequate socialisation levels were also only measured from the dogs’ reaction to a human experimenter in a very controlled environment, so may not be representative of the appropriate socialisation required to produce a dog that can cope in novel and potentially stressful situations [84,85,86]. Despite the socialisation protocols based on this early research, minimum levels of socialisation required to produce well-adjusted dogs have arguably still not been robustly investigated.

Fox and Stelzner (1966) [43] validated Scott and Fuller’s findings regarding the effects of the socialisation period on behaviour. They handled puppies from birth to 5 weeks and exposed them to different light, sound, and conspecific interactions. At 5 weeks, the puppies were observed for five minutes in an arena with a toy and cloth from the mother’s bedding, five minutes without these stimuli, and a further 5 minutes with the stimuli replaced. After this, a human approach test was carried out, followed by a detour task. Social behaviour of all puppies together in the arena was also recorded, as well as their behaviour with a human experimenter. Compared to control puppies (*n* = 8), handled puppies (*n* = 8) were hyperactive, showed higher levels of exploratory behaviour, were very sociable towards humans and were more dominant in social play with their littermates. They also performed best in the detour task, requiring fewer trials to pass around the barrier. Consistent with the findings of Scott and Marston (1950) [79], handled puppies also had a much higher heartrate than the control animals at five weeks, indicating the significance of experiences during this sensitive period. This study went further than the earlier research: as well as human handling, the treatment puppies were exposed to light and sound treatments, as well as exposure to water and an air jet (60 °F). However, the handling and other treatments were still done in a very controlled and clinical setting, not reflecting the average household stimuli that pet dogs are expected to cope with [43]. 

Defining the exact timing of the socialisation period in dogs is difficult due to breed-specific variation [27]. Morrow et al. (2015) [87] found that Cavalier King Charles Spaniel puppies had a significantly delayed onset of the early socialisation period compared to Yorkshire terrier puppies and German shepherd puppies. The exact timing of the socialisation period across all breeds has not been systematically tested. Therefore, as well as the timing, the content of socialisation and its effect on future behaviour should be a major point of focus [87].

These classic studies identified the existence and boundaries of the socialisation period, at least in some breeds [5]. Modern socialisation programs are largely based on Scott and Fuller’s research in the 1950s. These studies highlighted the importance of age-appropriate socialisation practices, with two main rules suggested for producing a “well-balanced and well-adjusted dog”. These rules were that 1) the ideal time for rehoming is 6- 8 weeks, and 2) young dogs should be introduced to the circumstances in which they will live as an adult before 3–4 months old. Although these are good socialisation rules to follow, the timing and content of a good socialisation protocol is still up for debate as the studies by Scott and Fuller, like much of the original research, are based on isolation experiments. This means that their value in telling us what is necessary for normal development is limited as we cannot distinguish the trauma caused by isolation, from the positive effects of human exposure. This means that for puppies who developed behavioural problems conclusions made regarding correct timings of human exposure cannot definitively explain the development of these behaviours, as they could have been due to the trauma of isolation. Future studies should focus on the wider developmental issues regarding the socialisation period rather than focusing on the ‘critical period’ concluded in these studies.

### 4.2. Socialisation Programs

Since the classic studies, very few have assessed the effects of full socialisation protocols on behaviour. We only identified three for this review. Kim et al. (2010) determined that socialised Jindo puppies (*n* = 6) exhibited a higher intensity of playful reactivity towards novel stimuli and a dog at 9 weeks compared to non-socialised puppies (*n* = 6) [70]. Vaterlaws-Whiteside and Hartmann (2017) [47] compared the behaviour of puppies exposed to a novel socialisation protocol (*n* = 19) with puppies experiencing the existing socialisation protocol (*n* = 14) in a guide dog facility at 6 weeks and 8 months. A Puppy Profiling Assessment (PPA; [62]) and Puppy Walking Questionnaire (PWQ; [63]) were used to assess puppies. Puppies that received the extra socialisation program had significantly more desirable scores in the PPA at 6 weeks. Puppies that received the new programme also had more favourable scores for separation-related behaviour, distraction, general anxiety, and body sensitivity at both 6 weeks and 8 months. The researchers tailored their socialisation protocol to coincide with the physiological and behavioural development of puppies as described by Scott and Fuller (1965) [27]. They also tested the puppies at 8 months of age, finding that early life socialisation positively impacts on later behaviour for traits, such as separation and general anxiety, body sensitivity, and distractibility [29,47].

Guide dog puppies provided intensive socialisation from birth to 8 weeks are more successful in training than puppies reared without this protocol [88]. In the third socialisation protocol study, Batt et al. (2008) [74] also hypothesized that training and socialisation would improve the success rates of dogs in the guide dog program. Interestingly, the treatments did not influence the success rate in guide dog training nor the likelihood of puppy raisers (i.e., people who raise guide dog puppies in their own home) raising a subsequent pup. The authors suggested that the high level of socialisation the puppies would have already received may explain the lack of significant difference in their groups. This indicates a maximum level of socialisation necessary to achieve desirable behavioural outcomes [74].

A few studies have focused on the effects of specific socialisation stimuli on behaviour, although these have not been linked to adult behaviour traits. Chaloupková et al. (2018) [60] investigated only auditory socialisation techniques. They hypothesised that exposure to audio stimuli during early ontogeny would improve reactions to noise during the police test for selecting puppies. Puppies in the treatment group (*n* = 19) were played ordinary radio broadcasts for 20 minute periods three times a day. At 7 weeks, these puppies and a control group (*n* = 18) were exposed to a sudden noise caused by a shovel, noise when alone in a room, and loud distracting stimuli. Puppies in the noise treatment exhibited higher scores following the sudden shovel noise than the control dogs. However, no significant differences were found in the other tests, suggesting that other aspects of the puppies’ rearing influenced their ability to cope with the other audio stimuli. This study highlights the importance of considering different sensory modalities when socialising a puppy [60].

In another study, Pluijmakers, Appleby, and Bradshaw (2010) [71] exposed puppies to audio-visual playbacks of stimuli, such as a vacuum cleaner, between the ages of 3–5 weeks. They then exposed the puppies to the real-life versions of the stimuli at 8 weeks of age. Puppies exposed to the audio-visual stimuli (*n* = 15) showed less fearful behaviour towards the real-life stimuli than control puppies (*n* = 13) [71]. This is an especially important study for shelter or breeding facilities, as exposure to audio-visual stimuli could bridge the gap between puppies’ maternal environment and the complex lives they are expected to cope with in the home. However, it is unclear whether the audio or visual aspect of the stimuli benefitted the puppies’ socialisation. 

Every study outlined in this section measured behaviour to assess the efficacy of a socialisation protocol. Behavioural tests and assessments are commonly used to decide whether working dogs are likely to be successful in further training (e.g., police dogs: [89]; guide dogs: [74,90]), to match shelter puppies with the right families, and to identify potential behavioural problems early. However, recent reviews have criticised the accuracy and reliability of existing assessments as very few assessments are developed using a systematic scientific approach and most lack reports of the test’s reliability and validity [91,92,93]. There is enormous variation in how tests are conducted, their application, the behaviour being assessed, and the dogs used, plus a range of other variables [92,94,95]. Going forward, great care needs to be taken in the application and validation of behavioural assessments. 

### 4.3. Questionnaires

Of the 28 studies included in this review, 18 rely on some type of survey or questionnaire. Several used validated surveys (e.g., C-BARQ [80]), but most research groups created novel questionnaires. The robustness of questionnaire-based studies to infer the effects of socialisation on behaviour has often been criticised [96]. These studies often rely on volunteer dog owners recruited by researchers. The outcome measures in turn may rely on reports from these volunteers, who will differ in their experience with dogs and subsequent perception of certain behaviours. Owners may also be biased about their own pet, potentially inflating the positive (or negative) outcomes of a socialisation treatment [97].

#### 4.3.1. Canine Behavioural Assessment and Research Questionnaire (C-BARQ)

The Canine Behavioural Assessment and Research Questionnaire (C-BARQ) is a list of 101 questions about how dogs respond to specific events and situations in their usual environment. The C-BARQ was designed to measure the prevalence and severity of behavioural problems in privately-owned and working dogs, and that remains its primary value and purpose [80]. However, Gonzalez-Martinez et al. (2019) [57] used the C-BARQ to assess the behavioural effects of six weeks of one-hour puppy classes, attended between 2 and 9 months of age, one year later. Both puppies and juveniles that attended classes had more favourable scores for family-dog aggression, trainability, non-social fear, and touch sensitivity [57]. This study demonstrates how the C-BARQ can be used to evaluate the outcomes of socialisation. However, the C-BARQ has not been validated for puppies, so it can only reliably assess adult behaviour.

Friedrich et al. (2019) [58] modified the C-BARQ to address socialisation, adding 15 questions relating to dog playfulness and removing 21 others for a total of 95 questions instead of the original 101. This study aimed to identify behavioural traits characteristic of German Shepherds, and to analyse the relationships between behavioural traits, and owner demographic and management factors. High scores for socialisation as a puppy were associated with lower scores for excitability and higher scores for stranger-directed interest and chasing [58].

Batt et al. (2009) [72] also modified the C-BARQ, with the aim of producing a questionnaire that related puppy raisers’ reports to guide dog performance. Out of the original 101 questions, 51 of them were used, as this study was for an Australian population, and many questions were deemed irrelevant. Puppy raisers’ predictions of success and the number of dogs in the household were the most important predictors of success in the guide dog training program. In relation to socialisation, the discussion of this study hypothesised that puppies in multi-dog households received less socialisation. This was explained by puppy raisers having less available time per dog, or a ‘paradoxical reduction’ in socialisation as the puppy raisers assume that puppies receive enough socialisation from the other dogs in the household [72]. 

#### 4.3.2. Non-C-BARQ

Many studies included in this review developed novel questionnaires to assess dog behaviour. The pros of these unique questionnaires are that they are specifically designed to assess the outcomes of socialisation, unlike the C-BARQ which gives a broad overview of behaviour, and they can be tailored to particular contexts, such as shelter environments or commercial breeding establishments. They also, similar to the C-BARQ, can amass huge datasets and include many breeds from different demographics. However, these questionnaires are not necessarily as well validated and are the reason this review has taken a narrative approach, as it is difficult to compare studies using very different questionnaires.

Many non-C-BARQ questionnaires are also completed by the owner. In an online survey of Finnish dog owners, Hakanen et al. (2020) [55] amassed data on 13,700 dogs with fears of specific stimuli. Dogs with a fear of fireworks, thunder, and novel situations had lower socialisation scores. The socialisation scores for this study were created from seven questions regarding experiences such as meeting strange people and travel into towns and cities between the age of 7 weeks and 4 months. All the questions had choices from never to several times a day and were scored from 0–5 (0, never; 1, 1–2 times during the puppyhood; 2, 1–2 times during the puppyhood to 2 times per month; 3, twice a month to twice a week; 4, twice a week to once a day; 5, several times a day). The scores were summed and thus the socialisation score varied between 0 and 35, with higher values indicating more socialisation events [55].In another owner-filled questionnaire, fearful dogs had fewer socialisation experiences defined by how much the dog met unknown women, men, children, dogs, visited the city, or travelled by car or bus, when the dog was between 8–12 weeks of age [65]. Another online study found that dogs aged between 2 months and 17 years with less socialisation during puppyhood were more likely to fear other dogs and strangers [56]. Moreover, Brand et al. (2022) [41] looked at the demographics and early life experiences of puppies acquired during the COVID-19 pandemic. Their findings, based on comparing 4369 “Pandemic Puppies” to 1148 “2019 puppies”, showed that Pandemic Puppies were significantly less likely to have attended and reaped the benefits of puppy training classes or been exposed to visitors to their home before the age of 16 weeks. This shift in socialisation has the potential to cause major welfare issues in Pandemic Puppies [41]. 

Some novel questionnaires were designed to assess puppies in the care of puppy handlers, or the effects of handler demographics on puppy behaviour. Mai et al. (2021) [53] found that puppy raisers who sought help on socialisation and training methods experienced better training outcomes for their puppies. Harvey et al. (2016) [63] used both an owner questionnaire and data about the home environment to assess factors influencing future guide dog behaviour. Dogs raised in a home with children scored higher on energy level, excitability, and distractibility. In addition, the more experience their carer had of walking puppies, the lower on energy level and distractibility subjects scored. The more puppies had been able to play with other dogs, the lower they scored on separation-related behaviour [63]. These findings show how environmental variables can affect behaviour, although the outcome measures are specific to desirable traits of guide dogs, so relevance to desirable companion animal behaviour is limited.

Many of these studies focus on specific behaviours, such as aggression or noise-related fear responses. Wormald et al. (2016) [98] assessed whether owners restricted their puppy to the household, if aggression and fear were present at an early age, and whether this affected the dog’s future behaviour. They found that there was actually no protective advantage of earlier or more frequent public exposure on the development of aggression as adults. This was demonstrated by survey results, which showed that every week an owner waited to expose their puppy to public areas, the more reduced the odds were that the puppy would show aggressive behaviour towards unfamiliar dogs as an adult. This study highlights the importance of more research being carried out on how dogs are socialised to unfamiliar dogs in early life, as this study demonstrates that mere simple introductions, i.e., exposures to unfamiliar dogs in a public park, may actually have more deleterious effects on future behaviour and their ability to cope with stress, when compared to more controlled, structured introductions during events like puppy classes [64,98]. However, Blackwell, Bradshaw, and Casey (2013) [67], who found that early exposure to particular noises were associated with fear of those noises as adults [67]. In a 2019 study, 14% of seized dogs that seriously wounded or killed another dog were likely to have had insufficient socialisation to other dogs during the socialisation period [59]. These studies suggest that aversive experiences with noises or other dogs may contribute to adverse reactions to these stimuli in later life, highlighting the need for positive interactions with potentially aversive stimuli in early life. However, similarly to all the questionnaire-based studies included in this review, this research only identifies correlations and not causations, again highlighting the need for more experimental, longitudinal studies [51].

Most questionnaire-based studies investigate either behaviour as a puppy or as an adult, with very few comparing the behaviour over time. One of the few longitudinal studies, Fuchs et al. (2005) [75], investigated the influence of external factors, such as socialisation, husbandry, and training on the results of a behaviour test and a questionnaire between 1 to 2 years of age and again one year later. Dogs from rescue shelters or with several previous owners obtained worse results in reaction to gunfire and “hardness” (severity or ability to accept unpleasant perceptions without being deeply impressed afterwards) than dogs that had come from homes or dogs that had few previous owners. The authors concluded that bad experience or lack of puppy training, socialization, and special training could have caused the effect, although this is speculation. Moreover, the testing was specifically designed to assess dogs for police work, so reactions to these test stimuli may not be relevant to companion animals [75].

Further studies have explored specific aspects of interspecific socialisation using questionnaires and testing. Arai, Ohtani, and Ohta (2011) [69] demonstrated how dogs’ contact with children during and after their socialisation period influences their future responses towards children. A questionnaire was used to ascertain levels of child exposure during the socialisation period. Treatment groups included dogs that had regular contact with children during the socialisation period (Group 1, *n* = 10), dogs that had occasional contact with children during the socialisation period (Group 2, *n* = 11), and dogs that only had contact with children after the socialisation period (Group 3, *n* = 10). Group 1 dogs showed no aggressive or excited behaviour towards the child during any intervention. Interventions for each group involved the child calling the dogs name repeatedly whilst standing in front of the door, walking directly towards the dog, and running around the dog calling his/her name. Group 2 dogs showed affinity behaviour, but also aggressive, escape, and excited behaviour when the child was active. Group 3 dogs showed a high level of aggressive behaviour, with a few showing affinity behaviour [69]. Dogs’ ability to behave appropriately towards children is crucial for child safety, and assessing their reactions to children is essential for matching shelter dogs to new owners. 

Finally, dogs’ ages when these questionnaires were filled out vary widely between studies. This is a potential issue as the longer it has been since the socialisation period, the more likely owners may be to forget or exaggerate aspects of socialisation. Future studies should aim to validate these questionnaires, and clarify the maximum age when socialisation can be accurately reported by owners.

### 4.4. Puppy Classes

Puppy classes are designed to safely and positively introduce puppies, with their new owners, to a range of stimuli (such as smells, sights, walking surfaces, items, unfamiliar people, and other dogs) during their sensitive period of socialisation [76]. There is some evidence that puppy classes positively influence adult behaviour, but other studies show no clear benefit [51]. Some authors even oppose socialisation before 16 weeks, due to the risk of contracting infectious diseases (e.g., parvo virus, as the final vaccination can be as late as 18 weeks of age) [99]. However, in one study, vaccinated puppies attending socialisation classes were at no greater risk of canine parvovirus infection than those that did not attend [100].

Cutler, Coe, and Niel (2017) [61] found that puppy classes improve behaviour after 20 weeks. According to owners, puppies that attended classes were less likely to fear noises (e.g., thunder) than puppies of non-attendees [61]. Casey et al. (2014) [66] demonstrated that attending puppy classes reduced aggression towards unfamiliar people [66]. When adopted from a humane society, dogs that participated in puppy classes were also more likely to be retained [76]. Gonzalez-Martinez et al. (2019) [57] used the C-BARQ to assess how six weeks of one-hour puppy classes affected adult behaviour. Puppies and juveniles that attended classes had better scores for family-dog aggression, trainability, non-social fear, and touch sensitivity [57]. Kutsumi (2012) [68] also used the C-BARQ, as well as a separate behavioural test, to assess the outcomes for puppies that attended either puppy classes (*n* = 44), puppy parties (*n* = 39), adult classes (*n* = 27), or no classes (*n* = 32). Attending puppy and adult classes were associated with higher responses to commands than the other two groups. Thus, participation in puppy and adult classes improved the obedience behaviour of dogs, regardless of age. The puppy class group had significantly more positive responses to strangers than the adult class and no class groups, and tended to have more favourable behaviour scores than the puppy party group [68]. Therefore, puppy classes may help to prevent canine behavioural problems, such as disobedience or fear of strangers.

However, Seksel, Mazurski, and Taylor (1999) [77] identified very few benefits of puppy classes. They found little significant difference between 5 treatment groups of puppies. Puppies had 4 weekly one-hour sessions of either: (1) both socialisation and training (*n* = 12), (2) socialisation only (*n* = 10), (3) training only (*n* = 13), (4) feeding only (*n* = 12), or (5) control conditions (*n* = 11). Although puppies in the socialisation and training and training-only groups received significantly higher ratings for their responses to commands at 2 and 4 weeks, there were no significant group effects on any other scales [77]. The range in ages of the puppies in each treatment group may be why few significant differences were observed between the groups. Given what we have discussed regarding the importance of timing of the sensitive periods of development, the benefits gleaned from the classes by a puppy at 6 weeks of age may have been very different from those experienced by a puppy at 17 weeks of age, which were the upper and lower age ranges of this study. A more powerful comparison between the groups could have been shown if all puppies underwent treatment at the same age, and although this author recognises the challenges of applied settings, future studies would benefit from testing puppies of the same age or correcting for age differences between groups [77].

Denenberg and Landsberg (2008) [73] evaluated the effectiveness of Dog Appeasement Pheromone (DAP) in reducing fear and anxiety in puppies and its effects on training and socialisation. Data from follow-up telephone surveys indicated that puppies in the DAP groups (*n* = 24) were better socialised and adapted to new situations and environments faster than puppies in the placebo groups (*n* = 21). Dogs in DAP and placebo groups displayed significantly reduced degrees of fear and anxiety. The DAP groups also exhibited longer and more positive interactions between puppies, including play [73]. Although this is interesting for general socialisation practises, the effects of the puppy classes themselves cannot be inferred from this study. However, given the mixed outcomes of the puppy class studies, DAP may make them more beneficial to puppies’ development, by reducing fear and anxiety.

To our knowledge, every study investigating the effects of puppy classes on adult behaviour used an owner-filled questionnaire to assess the outcomes. Variable experience, expertise, and bias are, therefore, potential issues. Some studies also tested behaviour, but the correspondence between these two methods were often not explored [101].

## 5. Future Directions

This review highlights that the majority of studies have used an observational questionnaire-based approach to examine this topic. There is a need for future studies that experimentally manipulate aspects of socialisation to enable the underlying causal mechanisms and outcomes to be quantified. The discussed studies highlight the importance of separate aspects of socialisation, such as auditory and visual stimulation (e.g., [60,71]). Individual studies can inform full socialisation protocols, and future studies could focus on the effects of specific stimuli, including different surfaces, toys, and the dog’s ability to generalise their reactions to related stimuli, such as people of different ages and ethnicities.

Age-appropriate socialisation is also critical and should inform future socialisation practices. Puppies should not be overwhelmed during the socialisation period, so adjusting stimulation to the level of development is a useful way to minimise the chances of this happening [47,51,102]. Future studies should focus on age-appropriate socialisation programmes that provide maximum benefits to adult behaviour.

Although not directly related to socialisation programs, the raising environment has profound effects on puppies’ future behaviour. Majecka et al. (2020) [103] found that outdoor-kennelled puppies showed more submissive behaviours, were more likely to show aggression through fear, and had less capacity for coping with novel situations [103]. Appleby et al. (2002) [104] found that dogs sourced from non-domestic maternal environments were more likely, during a veterinary examination, to show aggression towards unfamiliar people and avoidance behaviour [104]. In two investigations into Belgian dog breeders, a significant percentage of puppies never left their pen, encountered novel stimuli or unfamiliar people. Conditions seemed worse in the larger facilities. Source of acquisition can affect the incidence of separation-related behaviours, aggression, social and non-social fear, and touch sensitivity [30,66,105,106,107,108,109,110]. With the increase of large-scale breeding facilities globally, more applied research is required to assess the effects of these environments on dog behaviour and welfare [111].

Moreover, a major limitation of socialisation studies is that, in the real world, not all puppies are raised in the same environments [47]. This makes the outcomes of novel socialisation programs hard to assess, as many variables cannot be controlled. Accessing facilities that raise puppies in near identical environments or setting up dedicated labs can be difficult and costly. This is one reason why Scott and Fuller are still so widely cited, as subsequent large-scale investigations into early life experiences have not been carried out. However, the rise in facilities, such as commercial breeding establishments, may facilitate such studies in a semi-controlled setting. Dogs in these facilities are also typically intended as companion animals, so additional socialisation and assessments could focus on behaviours desirable in a pet, rather than a working dog. Working in existing facilities also reduces ethical concerns, because additional socialisation would only aim to improve puppy welfare, with controls experiencing baseline conditions stipulated by current legislation [26].

The maximum and minimum levels of socialisation required to produce emotionally robust adult dogs are currently unknown. This knowledge could inform legislation regarding levels of socialisation, and protocols could be adapted for breeding facilities to create the most cost-effective protocol that adequately prepares puppies for the home environment [51,87].

More research is needed to understand breed differences in the timing and requirements of socialisation. Persson (2018) [112] identified specific sociality genes associated with human-directed social behaviour in Golden and Labrador Retriever dogs. Such genes may influence the levels of socialisation required to produce well-adjusted adult dogs in these breeds, compared to other breeds [113]. Scott and Fuller also recognised that genetic differences affect the speed of development in different breeds, with some giant breeds taking considerably longer to reach developmental stages than toy breeds. This has serious implications for the implementation of age-appropriate socialisation, as well as the homing age for different breeds. At present, the most accepted time for rehoming puppies is at 8 weeks of age, but Serpell, Duffy, and Jagoe (2016) [26] state that this is a potential misinterpretation of current evidence [26]. More studies should be carried out demonstrating the effects of natural weaning on adult dog behaviour. 

Whilst this review has focused on socialisation in the form of exposure to interspecific social interactions and non-social stimuli, another important aspect of socialisation is intraspecific socialisation with the mother and littermates [114,115,116,117]. Previous studies have examined the timing of weaning and separation of puppies from the litter, and the current accepted time for rehoming stands at 8 weeks of age. This timeframe allows the puppy to reach its new home during the socialisation period, and interact with the stimuli it will have to cope with in its daily life. However, abruptly separating puppies at this age may cause acute stress and disadvantage the puppies’ development [87,116,117,118,119]. Pierantoni et al. (2011) [116] reported that puppies separated from the litter earlier (30–40 days of age) in the socialisation period were significantly more likely to exhibit problematic behaviours as adults than puppies adopted at 2 months [116]. Future research should focus on adequately socialising puppies during this period without subjecting them to the trauma of rehoming until they reach an age where they can more readily cope with this difficult transition.

One of the most important areas of future research should focus on teasing apart the developmental differences between ‘socialisation’ and ‘localisation’. As stated in the introduction, these terms have become synonymous in the literature, and although phenomenologically similar, they may represent different developmental processes. 

### 5.1. Assessing Outcomes

Standardised methods for assessing the behavioural outcomes of socialisation protocols are required, both to ensure that methods have been validated and to facilitate comparisons between them. The C-BARQ provides a promising candidate [80]. The Field Instantaneous Dog Observation tool has also been recently validated for adult dogs [120]. This is a quick, non-invasive test that can be accurately carried out by a range of people with varying experience of dog behaviour. This tool could be adapted to assess puppy behaviour and provide a standardised way to compare behaviour [120].

Future research should also focus on validating behavioural assessments. Valsecchi et al. (2011) [121] investigated inter- and intra-rater agreements, test-retest reliability, and validity of a temperament test for shelter dogs [121]. The behaviour assessment involved a battery of tests including exposure to aversive stimuli, such as an open umbrella, and puppies’ reaction to toys and food. To evaluate consistency in the behavioural assessment, dogs were tested twice in a shelter and once in their new homes 4 months after adoption. Performance in the test was generally consistent across time and between observers, although observers did not agree on some behaviours.

Few studies have explored the correspondence between questionnaire-based methods of behavioural assessment and other behavioural analyses. However, Barnard et al. (2012, 2017) [101,122] evaluated the consistency between a questionnaire-based method and behaviour of 2-month-old puppies in an open-field test, where various social and non-social stimuli were present. Correspondence between methods was high and test-retest consistency was also good using both evaluation methods [101,122].

Other cognitive tests could also be used to assess the outcomes of socialisation protocols. Problem-solving tasks (e.g., the impossible task paradigm and detour tasks) could test puppies’ problem-solving abilities and social cognition tasks (e.g., pointing/gazing tasks and social referencing tasks) could test their sociability. Both of these are invaluable traits for future training and bonding with a new owner [123,124,125,126,127,128,129,130,131,132,133,134,135,136,137]. Howell et al. (2011) review social cognition tasks as an easy form of socialisation for puppies, which breeders can carry out with minimal effort [138]. This combines both socialisation and testing to inform breeders of puppies’ general cognitive abilities and traits, potentially allowing breeders to match puppies to ideal owners.

To corroborate findings from behavioural assessments, future studies could also use physiological measurements. Oxytocin may be measured during socialisation to assess the benefits of certain methods of socialisation as it has been shown it increases affiliative behaviours in dogs, both towards humans and their conspecifics [139,140]. Stress measures could also be used to corroborate reports of fear-related behaviours from testing and surveys. Non-invasive indicators include salivary cortisol and faecal immunoglobulin A [141,142]. These measures provide empirical assessments of welfare that can be easily replicated and do not require experimenters trained in behaviour analysis to carry out the assessment.

Finally, more longitudinal studies are required to understand how early socialisation programs affect adult dog behaviour. Current evidence is not sufficient to infer whether the benefits of socialisation treatments continue into adulthood, or whether future poor conditions can counteract them.

### 5.2. Other Periods That Influence Adult Dog Behaviour

Although we have focused on primary socialisation, maternal care and even the prenatal period also affect animals’ future behaviour. For example, compared to rats from low-care mothers, rats from high-care mothers showed reduced fear responses, enhanced learning and memory, and reduced responses to stressors as adults [143]. Future studies on dogs could likewise investigate the influence of different levels of maternal care and the effects of prenatal stress on socialisation [144,145]. It is also unclear whether adequate socialisation can counteract poor experiences during these times [143,144,146]. Long-term epigenetic effects of poor experiences during these periods have also not been explored. Future research should focus on the epigenetic effects of these periods, as well as other important developmental periods [26].

It is hypothesised that early neurological stimulation, which provides episodes of mild early life stress, influences how well dogs cope with stress as adults. A practical example is daily handling, which is thought to have a positive impact on stress resilience, accelerates nervous system maturation, causes more rapid weight gain and hair growth, earlier opening of the eyes and enhanced development of problem-solving skills [102,147,148,149,150]. Although there have been previous studies focusing on the benefits of early neurological stimulation, recent studies have not been able to replicate the success of the original Bar Harbor study which was carried out by the U.S. military. The objective of what was called the Bio Sensor programme, was to create dogs that would have a superior advantage for military work. Future research should focus on validating this research.

Play is critical for development [151,152,153,154,155,156,157,158,159,160]. In dogs, play is thought to have three primary functions: locomotory development, training for the unexpected, and social cohesion. The process of socialisation overlaps with these primary functions, and therefore logically, socialisation should have an effect on play behaviour and vice versa. Future studies could investigate the influence of socialisation methods and content on the development of play in puppies and assess the outcomes in adult dogs [159].

Continuing socialisation after the primary period is also important for future behaviour [119,145,160]. Enrichment and proper stimulation later in life may partly compensate for a poor start, and later poor experiences can counteract an optimal early environment. However, there is little evidence to suggest whether counteracting the effects of poor early socialisation experiences in the late socialisation phase persist throughout adult life [29].

## 6. Conclusions

This review aimed to collate all studies regarding socialisation and demonstrate the gaps that still remain in our knowledge. The importance of socialisation cannot be understated and, as presented by this review, is a complex issue that has been addressed using a multitude of methodologies. With the ever-increasing population of pet dogs, now is a crucial time in the study of socialisation. More working relationships between researchers and breeders should be established with a hope to answer some of the outstanding questions in an applied setting. This will help eliminate any potential ethical questions regarding restriction of socialisation as any programmes will aim to improve welfare. This review strives to highlight the many avenues of research still to be pursued with a hope of improving the welfare of dogs globally.

## Figures and Tables

**Figure 1 animals-12-02895-f001:**
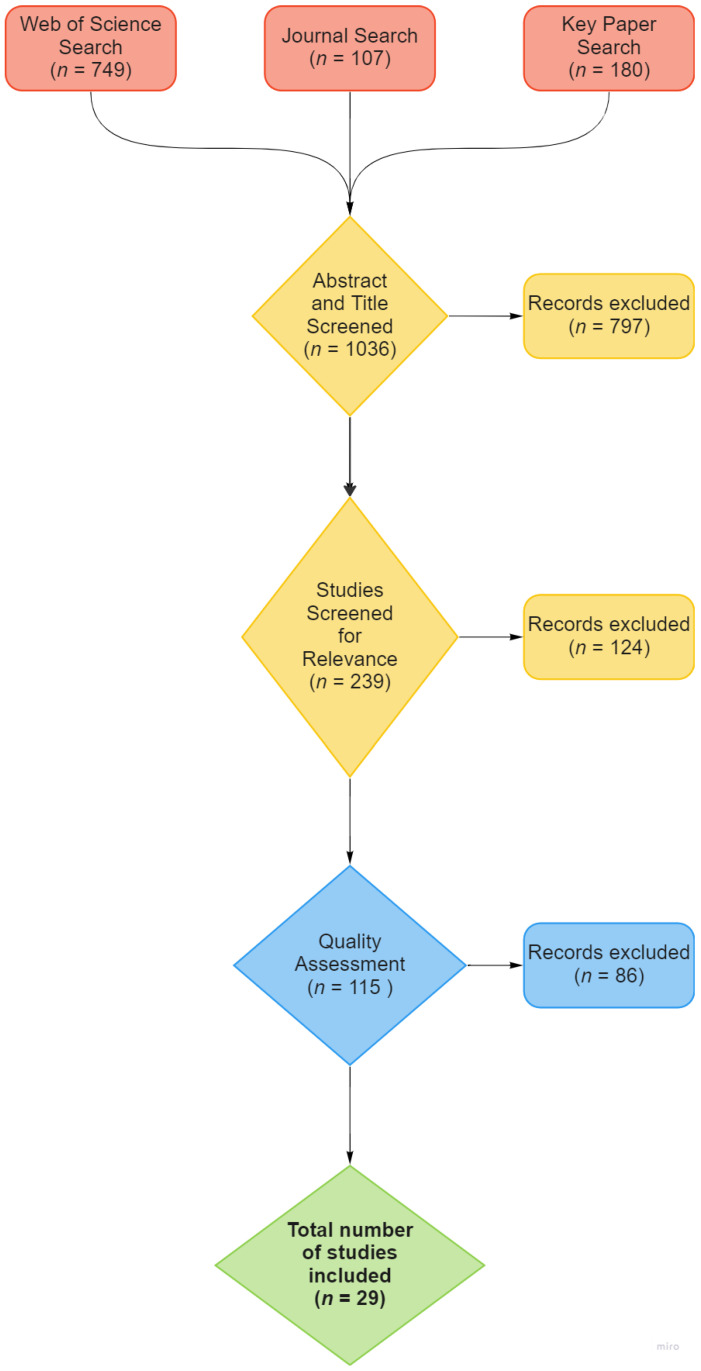
Flow diagram depicting search process.

**Table 1 animals-12-02895-t001:** All 29 studies that passed quality assessment to be included in main discussion.

Reference	Sample Size	Breed/Type of Dog	Age of Animals	Hypotheses/Aims/Objectives	Methods			Main Findings
					Treatment	Outcomes Assessed		
						Testing	Questionnaire	
(Brand et al., 2022) [41]	4369 (“Pandemic Puppies”) 1148 (“2019 Puppies”)	Various (Pet Dogs)	<16 wks ^4^	Explore impact of the 2020 COVID-19 pandemic on puppy early-life behaviour, socialisation/habituation experiences, and health.	X	X	Online, owner-completed survey; four sections including puppy demographics, health, behaviour and socialisation experiences.	Pandemic Puppies (aged <16 weeks) less likely than 2019 puppies to have attended puppy training classes or had visitors to their home.
(Mai et al., 2021) [53]	231 (Puppy raisers and dogs)	Various (Assistance Dogs)	3–25 mos ^1^	Investigate relationships between puppy raisers’ practices, provision of various supports to puppy raisers, and puppy behavioural outcomes.	X	X	Online, puppy raiser completed survey. Questions included demographic information and raiser practices, support and puppy behaviour using the Puppy Training Supervisor Questionnaire (PTSQ ^10^; Harvey et al., 2017) [54].	Puppy raisers that sought help for socialisation and training methods experienced better puppy outcomes.
(Hakanen et al., 2020) [55]	13,700 (9613 fear fireworks, 9513 fear thunder, 6945 fear novel situations, 2932 fear surfaces and heights.)	Various (Pet Dogs)	Survey: 2 mos ^1^ to 18 yrs ^2^	Identify environmental and demographic factors associated with non-social fearfulness.	X	X	Online, owner-completed questionnaire with background information and questions on seven canine anxiety traits.	Dogs that showed frequent non-social fear had experienced less socialisation during puppyhood.
(Puurunen et al., 2020) [56]	5973 (fear of dogs, non-fearful dogs, 4806 vs. fearful dogs, 1167) 5932 (fear of strangers, 896, vs. non-fearful dogs, 5036)	Various (Pet dogs)	Survey: 2 mos ^1^ to 17 yrs ^2^	Identify demographic and environmental factors associated with social fearfulness.	X	X	Online, owner filled. 7 main sections and demographic questions about socialisation.	Dogs with less socialisation during puppyhood more likely to fear other dogs and strangers.
(González-Martínez et al., 2019) [57]	80 (32 attended puppy classes, 48 did not.)	Various (Pet dogs)	Puppy classes: 2 to 9 mos ^1^ Survey: ≥15 mos ^1^	Determine effect of puppy classes on behavioural problems in adult dogs.	1 hour per week of puppy classes over 6 weeks.	X	C-BARQ ^3^ one year after completion of puppy classes.	Both puppies and juveniles that attended classes had better scores for family-dog aggression, trainability, non-social fear, and touch sensitivity.
(Friedrich et al., 2019) [58]	1041 dogs	German Shepherds	Survey: >2 yrs ^2^	Identify behavioural traits characteristic of German Shepherds; analyse relation between behavioural traits and demographic/management factors including levels of socialisation received as a puppy.	X	X	C-BARQ ^3^ and lifestyle survey	High scores for socialisation as a puppy linked to lower scores for excitability and higher scores for stranger- directed interest and chasing.
(Schilder et al., 2019) [59]	128 seized dogs; of 151 referred dogs in a clinical setting	Various (56% American Staffordshire/pit pull terrier type)	Veterinary examination: 9 mos ^1^ to 14 yrs ^2^	Investigating behavioural characteristics of “dog-killing dogs” to identify causes and motivational backgrounds including levels of socialisation received during the primary socialisation period.	X	X	Data gathered during behavioural anamneses in a veterinary clinic of seized dogs.	Aggressive dogs had less socialisation than other types of dogs during the primary socialisation period.
(Chaloupková et al., 2018) [60]	37 puppies (Treatment group = 19, control = 18).	Police dog breeds (Working dogs)	Audio stimulation: 16 to 32 days.	Analyse the effects of audio stimuli during early life.	Ordinary radio broadcasts played three times a day for 20-minute periods.	Exposure to a sudden noise, noise when alone, and loud distracting stimuli.	X	Treatment group puppies responded with a higher score, i.e., more positively, to the sudden noise than the control dogs.
(Cutler, Coe and Niel, 2017) [61]	296	Various (Pet dogs)	Surveys: < 20 wks ^4^	Characterise owner-reported experience of puppies attending socialisation classes, and owners’ approaches to socialisation	Responses compared between owners that did and did not attend puppy classes.	X	Participants completed a survey at enrolment and again when puppies were 20 wks ^4^ of age.	Attendee puppies less likely than non-attendee puppies of to show signs of fear in response to noises.
(Vaterlaws-Whiteside and Hartmann, 2017) [47]	33 (19 new socialisation program, 14 standard socialisation)	Guide dog breeds	Socialisation: 0 to 5 wks ^4^, PPA ^5^: 6 wks ^4^, PWQ ^6^: 8 mos ^1^	Design and evaluate new, inexpensive socialisation program.	New socialisation programme vs. standard breeder socialisation programme.	X	Puppy Profiling Assessment (PPA ^5^; Asher et al., 2013) [62] Puppy Walking Questionnaire (PWQ ^6^; Harvey et al., 2016) [63]	Puppies receiving extra socialisation had significantly better scores for separation-related behaviour, distraction, general anxiety and body sensitivity at both 6 weeks and 8 months.
(Harvey et al., 2016) [63]	224	Guide dog breeds	PWQ ^6^: 5 and 8 mos ^1^ PWQ ^6^ and “Environmental Information” survey: 12 mos ^1^	Explore how dogs’ home rearing environment will influence behavioural development.	X	X	Puppy Walking Questionnaire (PWQ ^6^; Harvey et al., 2016) [63] 11-item “Environmental Information” survey.	Dogs scored higher in energy level, excitability, and distractibility if raised with children, lower on energy level and distractibility with experienced carer, and lower on separation-related behaviour with more play with other dogs.
(Wormald et al., 2016) [64]	783	Various (Pet dogs)	Survey: 1 to 3 yrs ^2^ Acquired as puppies: <10 wks ^4^	This retrospective questionnaire aimed to quantify the amount of and age at which pet dogs received early social exposure compared to the levels of interdog aggression.	X	X	Questionnaire with 5 sections: (1) dog background; (2) early environment; (3) social exposure experience; (4) current behaviour; (5) health.	Early exposure of puppies in public areas was negatively correlated with reduced inter-dog aggression in adult dogs.
(Tiira and Lohi, 2015) [65]	3264	Various (192 breeds)	Survey: 3 mos ^1^ to 15 yrs ^2^	Investigate environmental factors linked to fear-related behaviours	X	X	Validated owner-filled questionnaire.	Fearful dogs had less socialisation experiences.
(Casey et al., 2014) [66]	3897	Various (Pet dogs)	Survey: 6 mos ^1^ to 17 yrs ^2^	Estimate number of dogs showing aggression to people in three contexts (unfamiliar people on entering, or outside the house, and family members) Investigate risk factors for aggression.	X	X	Questionnaire with four sections.	Attendance at puppy classes reduced risk of aggression to unfamiliar people.
(Blackwell, Bradshaw and Casey, 2013) [67]	3897 (postal survey) 383 (structured interview)	Various (Pet dogs)	Survey: 6 to 216 mos ^1^	Investigate prevalence and characteristics of noise-associated fear; identify risk factors and any co-morbidity with separation-related behaviour and fear responses in other contexts.	X	X	Postal survey of dog owners to investigate general demographic factors, and structured interviews to gather more detailed information.	Early exposure to noises a risk factor for specific fears.
(Kutsumi, 2012) [68]	142 (44 Puppy classes, 39 puppy parties, 27 adult classes, 32 no classes)	31 breeds representing sporting, hound, terrier, toy, non-sporting, and herding demographics	Puppy classes: ~4 mos ^1^	Clarify whether puppy socialisation and command training class prevented behaviour problems in dogs.	Puppy classes 1 h each week for 6 weeks; puppy parties 1 h each week for six weeks; adult class involved obedience lessons for 1 h each week for 6 weeks; no class group underwent no formal training.	Behaviour test evaluating response to commands, owner’s recall, separation, a response to novel stimulus and strangers.	C-BARQ ^3^	Adult and puppy class dogs responded to commands better; puppy classes dogs had more positive responses to strangers.
(Arai, Ohtani and Ohta, 2011) [69]	31 (10 dogs had experience of children during and after socialisation period, 11 dogs had experience after only, 10 dogs had no experience)	Various (13 breeds, pet dogs)	Dogs were initially acquired between birth and 12 mos ^1^.	Demonstrate how dogs’ contact with children during and after socialisation period influenced responses toward children.	Dogs that had contact with children during socialisation period; dogs that had contact with children after the socialisation period; dogs that seldom had contact with children.	Exposure of dogs to a novel child exhibiting three behaviours including calling the dogs name repeatedly whilst standing in front of the door, approaching the dog and calling the dogs name whilst running around it.	Questionnaire to ascertain levels of child exposure during socialisation period.	Dogs that had contact with children during and after the socialisation period showed no aggressive or excited behaviour towards children. Dogs that only had contact with children after the socialisation period showed some affinity behaviour but also aggressive, escape and excited behaviour when the child was active. Dogs with no exposure to children showed lots of aggressive behaviour and little affinity behaviour.
(Kim et al., 2010) [70]	12 (6, Socialised and 6, non-socialised.)	Jindo (Laboratory dogs)	Treatment and baseline testing: 7 wks ^4^ Testing: 9, 11, 13 and 60 wks ^4^	Determine whether socialised puppies showed different behavioural reactivity from non- socialised puppies.	Puppies assigned to a socialised group or a non-socialised group.	5 behavioural tests.	X	Socialised Jindo puppies exhibited more intense playful reactivity towards novel stimuli and a dog at 9 weeks. There were no significant differences between the groups at 11, 13 or 60 weeks.
(Pluijmakers, Appleby and Bradshaw, 2010) [71]	Experiment 3: 28 (15 treatment, 13 Control)	Various commercially bred dogs (3 breeds) Maltese Terrier Boomer and Jack Russell Terrier.	Treatment: 3 to 5 wks ^4^ Testing: 51 to 61 days (Experiment 3)	Tested whether exposure to audio visual playback reduced fearful and increased exploratory behaviour	Experiment 3: Half of each litter 30 mins each day for 2 weeks, video footage and the other half acted as controls. Control: 30 mins each day for 2 weeks, blank screen	Testing at 36 days in a familiar environment and an unfamiliar environment with objects corresponding to those in the video footage and unfamiliar objects.	X	Puppies exposed to the video images were less fearful than the non-exposed puppies. The control puppies held their ears back between the partial and maximal position for significantly longer than the exposed puppies and also were more likely to exhibit a crouched posture.
(Batt et al., 2009) [72]	111	Guide dog breeds (Working dogs)	Survey: 13 mos ^1^	Design a questionnaire that related puppy raisers’ reports to guide dog performance.	X	X	Modified C-BARQ ^3^	Puppy raisers’ predictions of success and number of dogs in the household best predicted success in the guide dog training program.
(Denenberg and Landsberg, 2008) [73]	45 (24 DAP, 21 placebo)	2 large and 2 small breed groups (Laboratory dogs)	Puppy classes: 12 to 15 wks ^4^	Evaluate effectiveness of DAP ^8^ in reducing fear and anxiety, and effects on training and socialisation.	Four groups of puppies in puppy classes: 2 large-breed groups (1 DAP ^8^ and 1 placebo group) and 2 small-breed groups (1 DAP ^8^ and 1 placebo group).	X	Classes lasted 8 weeks; owners completed questionnaire before and after each lesson. Follow-up telephone surveys on subsequent socialisation of puppies 1, 3, 6, and 12 months after classes ended.	Dogs in DAP ^8^ groups less fearful and anxious than placebo groups; DAP ^8^ groups displayed longer and more positive interactions between puppies, including play.
(Batt et al., 2008) [74]	60 (20 training, 20 socialisation, 20 control)	Guide Dogs Breeds (Working dogs)	Treatment: 12 to 16 wks ^4^ Testing: 14 mos ^1^	Explore whether training and socialisation improve success rates in guide dog program.	Training, socialisation, and control.	Success in Guide Dog programme.	X	Socialisation/training treatments did not influence success rate nor likelihood of puppy raisers raising another pup.
(Fuchs et al., 2005) [75]	149	German Shepherds (Pet dogs)	1–2 yrs ^2^	Investigate influence of external factors like socialisation, husbandry, training on results of behaviour test that focused on seven traits, self confidence, nerve stability, reaction to gunfire, temperament, hardness, sharpness, defense drive, and overall behaviour and compare test consistency after one year.	X	30–40 min behavioural test exposing dogs to various stimuli (described in detail by Ruefenacht et al., (2002)).	Questionnaire covering husbandry, training, socialisation, and behaviour in certain situations, etc. before first test. After 1 year, 38 dogs tested again, alongside another, similar questionnaire.	Dogs from rescue shelters or with several previous owners received worse results in reaction to gunfire and “hardness”, which is defined as severity or ability to accept unpleasant perceptions without being deeply impressed afterwards.
(Duxbury et al., 2003) [76]	248 (87, Humane Society socialisation classes, other socialisation classes, 132, no socialisation classes, 29)	Not specified Pet dogs	Treatment: 7 to 12 wks ^4^ Survey: 1 to 6.5 yrs ^2^	Identify associations between retention of dogs in their adoptive homes and attendance at puppy socialisation classes (and other factors)	Puppies either underwent socialisation classes or did not.	X	Epidemiologic survey on adult dogs that were adopted as puppies from a humane society.	Higher retention for dogs that participated in Humane Society socialisation classes and were handled frequently as puppies.
(Seksel, Mazurski and Taylor, 1999) [77]	58 (12, Socialisation plus Training S/T, 10, Socialisation, 13, Training, 12, Feeding and 11, Control)	Various (36 breeds) Pet dogs	Treatment and Testing: 6 to 16 wks ^4^ Survey: 4 to 6 mos ^1^ following the completion of the program, and before start of program.	Puppies that underwent socialisation hypothesised to be better behaved, score higher in the handling, social stimuli, and novel stimuli category.	S/T puppies attended Puppy Preschool class for 1 h; Training group received 10 mins ^9^ training per week; Socialisation group received only socialisation experiences; Feeding group given treats equal other groups; Control group attended the veterinary clinic for 15 mins ^9^ (All for 4 wks ^4^)	Battery of tests scored by four scales of social, novel, handling, and commands scores.	X	Puppies in the S/T and training groups received significantly higher ratings for their responses to commands at 2 and 4 weeks into the programme. No significant group effects on any other time-scales.
(Fox and Stelzner, 1966) [43]	22 (8 control, 8 handled, and 6 partially socially isolated)	Not specified Laboratory dogs	Treatment: Birth to 5 wks ^4^ Testing: 5 wks ^4^	Determine the effects of differential rearing on behaviour and development.	Handling carried out from one day until 5 wks ^4^ of age. Handling included light, sound and conspecific interactions.	Arena test Approach test Detour test	X	Handled puppies hyperactive, more exploratory, very sociable towards humans, and more dominant in social play. They also preformed best in the detour task.
(Freedman, King and Elliot, 1961) [78]	34 (6 two weeks, 6 three weeks, 7 five weeks, 7 seven weeks, 3 nine weeks, 5 controls)	Cocker spaniels and beagles Laboratory dogs	Treatment: 2 to 14 wks ^4^ Testing: 14 wks ^4^	Identify age when human contact most reduces withdrawal response at 14 wks ^4^	Puppies taken for a week of socialisation at 2, 3, 5, 7 and 9 wks ^4^ of age. Controls remained in the field.	Handling test at 14 wks ^4^.	X	Puppies increasingly withdrew from humans if taken for socialisation after 5 wks ^4^ of age. If taken after 14 wks ^4^, normal human relationships could not be established.
(Pfaffenberger and Scott, 1959) [42]	154 (40, 0–1 wks ^4^ in kennel. 22, 1–2 wks ^4^. 18, 2–3 wks ^4^, 3 or more wks ^4^. 30, controls and dogs which failed the initial puppy testing program from 8–12 wks ^4^. 124, puppies which passed the initial testing.)	Various (4 breeds) Guide dogs Working dogs	Treatment: 12 to 23 wks ^4^	Identify factors affecting success rates of guide dogs.	Rehomed at 12 wks ^4^ or spent longer at kennel before rehoming (1–11 wks ^4^).	Success in guide dog training.	X	Dogs homed after 12 weeks passed training with approximate 90% success rate. Dogs placed in second week after the 12 weeks performed slightly poorer, but not significantly so. Dogs retained in kennel more than two wks ^4^ had more failures.
(Scott and Marston, 1950) [79]	73 (20 observational data only, 53 both test and observational data)	Basenji, Beagle, Cocker spaniel, Dachshund, Shetland Sheep Dog and Wire-haired Fox Terrier.	Testing: 5, 7, 9, 11, 13, and 15 wks ^4^.	Study whether development of social behaviour in puppies occurs during critical periods when experiences have long-lasting effects.	X	Testing included relationships with handlers, dominance, confidence-timidity rating, activity ratings, changes in heart rate and body weight.	X	Disturbances during development are most important during periods when new social relationships are being formed. Also detailed critical periods of dog development.

^1^ Mos: Months. ^2^ Yrs: Years. ^3^ Definition of C-BARQ: The Canine Behavioural Assessment and Research Questionnaire is an owner questionnaire consisting of 101 questions related to the (1) training and obedience, (2) aggression, (3) fear and anxiety, (4) separation-related behaviour, (5) excitability, (6) attachment and attention seeking, and (7) miscellaneous behaviours, e.g., chasing, urination of their dog (C-BARQ; Hsu and Serpell, 2003) [80]. ^4^ Wks: Weeks. ^5^ Definition of Puppy Profiling Assessment: (PPA; Asher et al., 2013) [62]. ^6^ Definition of Puppy Walking Questionnaire: (PWQ; Harvey et al., 2016) [63]. ^7^ NSW/ACT: New South Wales/Australian Capital Territory. ^8^ DAP: Dog appeasing pheromone. ^9^ Mins: Minutes. ^10^ Definition of the Puppy Training Supervisor Questionnaire (PTSQ; Harvey et al., 2017) [41]. X. It is to indicate that a respective study did not include the relevant methodologies.

## Data Availability

Not applicable.

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
