# Peer review of "Canine Socialisation: A Narrative Systematic Review"

_animals, 2022, doi:10.3390/ani12212895_

Round 1

Reviewer 1 Report

This has clearly been a substantial undertaking and I congratulate the authors on it.  I enjoyed reading it, and have a few comments which I hope are helpful.

As the authors are no doubt aware, the work of Scott and Fuller was based on isolation studies and accordingly their value in telling us what is necessary for normal development is limited as we cannot distinguish the trauma caused by isolation and / or exposure to others at different ages from the positive effects of exposure. Thus these experiments can only really tell us what is not necessary for normal development; i.e. the only real scientific insight comes from the puppies that developed normally, - for these we can say with confident the isolation / deprivation had no effect. For all who developed problems it is much harder to tease out the reasons why, despite the common conclusion about critical periods.  This point seems to have been missed by the authors and I think deserves discussion, since there is a danger that this field has a strong confirmation bias with studies seeking to reinforce the initial finding rather than considering the wider developmental issues.

A second point that I think deserves comment, is the (reasonable) decision to exclude studies with a sample size of lass than 5  animals  (l153). I fully support this, but in the wider discussion I think it would be valuable to highlight how such a criterion would exclude several of the studies by Scott and Fuller and Freedman, e.g. the seminal study of  Freedman, D. G., King, J. A., & Elliot, O. (1961). Critical period in the social development of dogs. Science, 133(3457), 1016-101; for which the 9 week group are just 3 puppies. Thus to also say that this is definitive (l226) is perhaps a stress. Indeed the failure to socialise the completely isolated puppy is n=1.

I could not work out why certain cells were coloured in Table 1. Please could you explain?

As I read the discussion I began to question whether my definition of socialisation was the same as the authors. I think it is essential that the authors clearly define what they mean by this term and stick to it throughout. To me, socialisation has 2 components- either of which might be defined as socialisation. 1 accepting other social stimuli. 2. Learning the social skills necessary to fit in with a social group. However in the discussion the authors seem to extend it to more general habituation to non-social stimuli (l249). This probably involves a different developmental process even if it is phenomenologically similar to type 1 socialisation described above. I think the script would benefit from clarifying these distinctions.  

Minor points

Line 65-67- it seems somewhat circular to reason that dogs becoming aggressive is the main reason for aggressive behaviour towards humans.  Please rephrase these 2 sentences.

In the discussion, where the authors discuss specific studies eg l338ff it would be useful to have the n for the various groups.

In the section where they refer to the potential to investigate quality of maternal care, I think it would be useful to reference the work of Foyer who has published on this.

Reviewer 2 Report

The manuscript provides comprehensive coverage of socialization in dogs, however there are some minor changes that need to be addressed.

L 50: critical and sensitive are not synonymous

L 67: I don´t understand. You mean aggression is the main reason pet dogs are aggressive?

L 234-236: The sentence is not clear, rewrite it please.

L 240: Not clear which is the best timing of human socialization.

L 338: Arai, Ohtani 338 and Ohta (2011). Is a socialization protocol applied in this study? not sure to put it in this section

L 401-403: More details about this study would be interesting.

L 418-419: It would be useful to have more details about the socialization scores.

L 503-509: What is your hypothesis/ opinion about the lack of correspondence between these results and those cited above where beneficial effects of the puppy classes were observed?

L 661-666: The relationship between this paragraph and the objective of the study is not clear

Reviewer 3 Report

The paper is a complete and in-depth review on canine socialization. This aspect, very important for the development of a correct social behavior in the dog, has not been addressed in a univocal and complete way in the past. This review aims to analyze the research carried out on this topic, selecting them on the basis of strict scientific criteria. This selection led to the identification of 29 papers out of 115.

The discussion analyzes in detail the individual aspects dealt with in the 29 papers, with an in-depth analysis, rich in bibliographical references.

In my opinion the review is the result of an excellent research and analysis work that will allow the reader to have a complete and updated picture of all the research carried out in this field.

The paper can be accepted for publication and I congratulate the authors for the excellent work they have done.
